# Gender and Socioeconomic Differences in the Prevalence and Patterns of Multimorbidity among Middle-Aged and Older Adults in China

**DOI:** 10.3390/ijerph192416956

**Published:** 2022-12-16

**Authors:** Yaqin Zhong, Hanqing Xi, Xiaojun Guo, Tiantian Wang, Yanan Wang, Jian Wang

**Affiliations:** 1School of Public Health, Nantong University, Nantong 226019, China; 2School of Medicine, Nantong University, Nantong 226019, China; 3School of Science, Nantong University, Nantong 226019, China; 4Dong Fureng Institute of Economic and Social Development, Wuhan University, Wuhan 430072, China

**Keywords:** multimorbidity, patterns, gender differences, socioeconomic difference

## Abstract

Background: Multimorbidity has become a global public health concern. Knowledge about the prevalence and patterns of multimorbidity will provide essential information for public intervention and clinical management. This study aimed to examine gender and socioeconomic differences in the prevalence and patterns of multimorbidity among a nationally representative sample of middle-aged and older Chinese individuals. Methods: Data were obtained from the 2018 wave of the China Health and Retirement Longitudinal Study. Latent class analysis was conducted to discriminate among the multimorbidity patterns. Multinomial logit analysis was performed to explore gender and socioeconomic factors associated with various multimorbidity patterns. Results: A total of 19,559 respondents over 45 years old were included in the study. The findings showed that 56.73% of the respondents reported multimorbidity, with significantly higher proportions among women. Four patterns, namely “*relatively healthy class*”, “*respiratory class*”, “*stomach-arthritis class*” and “*vascular class*”, were identified. The women were more likely to be in the *stomach-arthritis class.* Respondents with a higher SES, including higher education, urban residence, higher consumption, and medical insurance, had a higher probability of being in the *vascular class.* Conclusions: Significant gender and socioeconomic differences were observed in the prevalence and patterns of multimorbidity. The examination of gender and socioeconomic differences for multimorbidity patterns has great implications for clinical practice and health policy. The results may provide insights to aid in the management of multimorbidity patients and improve health resource allocation.

## 1. Introduction

Multimorbidity can be defined as the coexistence of two or more chronic diseases, and it has become a global public health concern [1]. Multimorbidity is associated with impaired functional ability, higher healthcare costs, poorer quality of life and an increased mortality risk [2,3], challenging healthcare systems around the world. Dealing with multimorbidity is complex for both healthcare systems and research [4,5].

Because of the large heterogeneity in terms of definitions, samples and methods among studies, the prevalence of multimorbidity varies across countries and districts. According to the World Bank, the prevalence of multimorbidity is 26.2% for sub-Saharan Africa, 29.5% for Asia, 31.8% for East Asia, 33.1% for the Middle-East and North Africa, 44% for Europe and Central Asia and 50.4%% for Latin America and the Caribbean [6]. A systematic review indicated that the prevalence of multimorbidity ranged from 6.4% to 76.5% among adults aged 60 years or more in China [7].

Major predictors for multimorbidity include individual factors, socioeconomic status, health behaviors and environmental factors. Some studies found that women had significantly higher odds of multimorbidity compared with men [8,9,10], whereas others showed a non-significant association [11]. A review assessed the association between education and multimorbidity in 31 studies, and most studies reported that the risk of multimorbidity was higher among individuals with lower education, while some studies reported a lower risk with lower education [6]. Some studies found an association with a higher prevalence of multimorbidity for people in the most well-off class, while others stated that the prevalence was higher for people considered to be poor [8,10]. There were significantly higher risks for multimorbidity in urban areas [12,13].

Previous research has found that chronic diseases tend to cluster together into so-called multimorbidity patterns [3]. Knowledge about the patterns of multimorbidity will provide essential information for public intervention and clinical management [4]. However, multimorbidity patterns in different countries and regions present differently due to different populations and methodologies [14]. A study conducted among Swedish older adults identified six patterns of multimorbidity: (i) psychiatric disorders; (ii) cardiovascular diseases, anemia and dementia; (iii) metabolic and sleep disorders; (iv) sensory impairments and cancer; (v) musculoskeletal, respiratory and gastrointestinal diseases; and (vi) unspecific. Garin et al. analyzed data from the Collaborative Research on Ageing in Europe project and the World Health Organization’s Study on Global Ageing and Adult Health and identified a “cardiorespiratory pattern”, “metabolic pattern” and “mental-articular pattern” [9]. A study conducted in Shanxi province in China found that “degenerative/digestive diseases”, “metabolic diseases” and “cardiovascular diseases” were the three specific multimorbidity patterns [1]. Previous findings of multimorbidity patterns may not be comparable due to differences in the study population, eligible diseases and analytical methods [15].

The applicability of multimorbidity patterns requires further knowledge of the prevalence, the chronic conditions that are involved, the risk factors and the existence of potential gender and socioeconomic differences [16]. A deeper understanding of the gender and socioeconomic differences in multimorbidity may lead to preventive actions to diminish the prevalence and give rise to new, comprehensive approaches for the management of these co-occurring diseases. Therefore, the present study was undertaken to examine the gender and socioeconomic differences in the prevalence and patterns of multimorbidity among middle-aged and older Chinese individuals.

## 2. Methods

### 2.1. Data and Sample

Data came from the 2018 wave of the China Health and Retirement Longitudinal Study (CHARLS). A national representative survey of China’s middle-aged and older adults, CHARLS comprises a three-stage stratified probability proportionate to size (PPS) sample. CHARLS covers 28 provinces, 150 counties or districts and 450 urban communities or villages across the country. Details of CHARLS have been described elsewhere [17,18]. Respondents with missing values for multimorbidity and other important variables were excluded from analyses, resulting in a total of 19,559 respondents in the present study. All analyses were weighted using individual sample weights, adjusting for non-responses of individuals and households.

### 2.2. Variables and Instruments

#### 2.2.1. Multimorbidity and Multimorbidity Patterns

In CHARLS, fourteen self-reported chronic diseases are assessed by asking “Have you been diagnosed with the following conditions by a doctor: hypertension, diabetes, dyslipidemia, cancer, liver diseases, chronic lung diseases, heart diseases, stroke, digestive diseases, kidney diseases, memory-related disease, emotional or psychiatric problems, arthritis and asthma?”. All diseases and conditions are defined as a binary variable (yes vs. no). Multimorbidity is defined as the simultaneous presence of two or more chronic diseases/conditions within one person. Older adults with two or more co-existing diseases/conditions were selected to perform an exploratory latent class analysis (LCA), and we clustered them into different latent class groups.

#### 2.2.2. Socioeconomic Status

Socioeconomic status (SES) was indicated by the following measures: education, residence, In per capita expenditure (In PCE) and types of medical insurance. Education is a well-known measure for SES. In CHARLS, education is obtained from the question, “What is the highest level of education you have completed?” Education is categorized into four levels: (1) “Sishu/home school and below”, including those who can neither read nor write, those who were reported to have been in “Sishu”, or those who did not finish primary school; (2) “elementary school”, including those who have completed primary school; (3) “middle school”, including those who have completed a junior-high-school-level education; (4) “high school and above”, including those who have completed a senior high school, vocational school, college or graduate-level education. Another measure of SES was In PCE. In CHARLS, household expenditures are collected weekly, monthly and yearly to minimize recall bias. Previous literature has shown that in developing countries, expenditure provides a better measure than wealth or income [19]. Due to differences in urban and rural economic development levels and household registration system designs in China, there are significant differences in income, social resource allocation and access to welfare policies. A previous study has documented gaps and trends in health disparities between urban and rural areas [20,21]. Thus, residence was used as another proxy measure for SES. In the 2018 CHARLS survey, there were three main types of social medical insurance in China: urban employee basic medical insurance (UEBMI), urban resident basic medical insurance (URBMI) and the new rural cooperative medical scheme (NRCMS) [22]. Therefore, in the present study the types of health insurance included no health insurance, UEBMI, URBMI and NRCMS and others (private medical insurance and other medical insurance).

#### 2.2.3. Covariates

Demographic variables included age, gender (male, female) and marriage status (married, windowed, other). The health risk factors consisted of smoking status (never, quit, current) and alcohol consumption (never, occasionally, usually). The health-related variables included difficulty in activities of daily life (ADL: yes, no) and self-rated health (SRH: less than good, good). Items of ADL included dressing, bathing/showering, eating, getting into and out of bed, using the toilet and bladder and bowel control. For each item, respondents indicated their level of difficulty in performing the activity [23]. Two dichotomous variables (yes or no) were used to indicate whether respondents had difficulty in ADL.

### 2.3. Statistical Analysis

Descriptive statistics were conducted to summarize overall information. Categorical variables were expressed as frequencies (percentage) and continuous as means (standard deviation, SD). The chi-square test was used to assess differences between multimorbidity and sociodemographic characteristics.

LCA was performed to identify the clustering patterns of 14 different chronic conditions or diseases among the 19,559 respondents. LCA is based on structural equation modeling and is useful in determining subtypes of cases or groups in multivariate categorical data [1]. It is suitable for describing clusters of respondents according to their chronic disease patterns [24]. The adjusted Bayesian Information Criterion (BIC) and the consistent Akaike Information Criterion (AIC) were used to determine the optimal number of latent classes. Based on the evaluation of a variety of model fit statistics, the present study examined two to six classes and selected the best-fitting solution.

Multinomial logit analysis was conducted to explore the association between multimorbidity patterns and gender and SES, while accounting for other potential confounding variables as covariates (age, education, ADL, SRH, smoking status and alcohol consumption). Statistical analyses were performed using Mplus 8.3 and Stata 14.0. Two-sided *p* values below 0.05 were considered statistically significant.

## 3. Results

### 3.1. Sample Characteristics

Descriptive characteristics of the sample and the prevalence of multimorbidity are presented in Table 1. A total of 19,559 respondents over 45 years old participated in the study; 52.42% of them were women. Moreover, 43.65% of the respondents had a Sishu/home school education and below; 59.69% of them lived in rural areas. For health insurance, respondents with no insurance, UEBMI, URBMI, NRCMS and others accounted for 2.98%, 15.21%, 16.30%, 64.41% and 1.09%, respectively. The mean number of chronic diseases for all participants was 2.17, and it was 2.08 for males and 2.26 for females, respectively. Overall, 56.73% of the respondents had two or more chronic conditions simultaneously, this rate being even higher in women (58.71%) and in those with a Sishu/home school education and below (60.49%). Respondents who never consumed alcohol, had quit smoking and had difficulty in ADL were more likely to display multimorbidity.

### 3.2. Gender Differences in Multimorbidity

Figure 1 displays the weighted prevalence of multimorbidity by age and gender. The prevalence of multimorbidity increased from 34.20% in the respondents aged 45–50 years to 60.38% in those aged 61–65 years and 70.30% in those aged 76–80 years, and then slightly decreased to 66.67% in respondents aged more than 80 years old. In general, women had a higher prevalence of multimorbidity than men (Figure 1). Similar trends were found in the number of chronic diseases by age and gender (Figure 2).

### 3.3. Multimorbidity Patterns

In this study, we used LCA to examine two to six class models and identified the clustering of diseases. By comparing the Bayesian Information Criterion (BIC), the *p* value of the Bootstrap Likelihood Ratio Test (BLRT) and the interpretability of each class model, the four-class model emerged as the best-fitting one [14]. More detailed information is included in the Appendix A. All 19,559 respondents were classified into one of the four classes, which were named as the *relatively healthy class*, *respiratory class*, *stomach-arthritis class* and *vascular class*. The *relatively healthy class* was composed of respondents with a substantially lower prevalence of all chronic diseases. The *respiratory class* included respondents with a higher prevalence of lung disease and asthma. The *stomach-arthritis* class consisted of respondents with a higher prevalence of stomach diseases and arthritis. The *vascular class* included respondents with a higher prevalence of hypertension, dyslipidemia, stroke and heart disease. As shown in Appendix A, 65.02% of the respondents belonged to the *relatively healthy class*. Around 6.39%, 9.99% and 18.60% of the respondents were assigned to the *respiratory class*, *stomach-arthritis class* and *vascular class*, respectively.

### 3.4. Gender and Socioeconomic Differences in the Patterns of Multimorbidity

Table 2 presents the results of the associated factors of multimorbidity patterns, reporting the relative risk ratio (RRR) and 95% CI. The females had a higher probability of being in the *stomach-arthritis class* (compared with the relatively healthy class), with an RRR of 1.595 (95%CI: 1.276–1.993). As age increased, respondents were more likely to be classified into the *respiratory class* (RRR = 1.038, 95%CI: 1.026–1.050), *stomach-arthritis class* (RRR = 1.030, 95%CI: 1.023–1.038) and *vascular class* (RRR = 1.033, 95%CI: 1.027–1.041). Regarding alcohol consumption, respondents who occasionally or usually drank had a lower possibility of being in the respiratory class and stomach-arthritis class. Compared to those who never smoked, respondents who currently smoked or had quit smoking were more likely to be in the *stomach-arthritis class*.

With respect to socioeconomic factors, we observed a positive association between higher education, urban residence and higher consumption with the *vascular class*. Compared with the relatively healthy class, respondents with a high school education and above had a higher possibility of being in the *vascular class* (RRR = 1.363 for middle school and 1.401 for high school and above). Respondents living in an urban residence were 1.438 times (95%CI: 1.236–1.672) more likely to be classified into the *vascular class.* Respondents with higher consumption were associated with an increased likelihood of being in the *stomach-arthritis class* (RRR = 1.095, 95%CI: 1.026–1.168) and *vascular class* (RRR = 1.114, 95%CI: 1.053–1.179). Compared to the case of no health insurance, all types of health insurance were associated with higher probabilities of being in the *vascular class*, with an RRR of 3.200 (95%CI: 2.148–4.768), 2.353 (95%CI: 1.608–3.443), 2.021 (95%CI: 1.403–2.912), 2.385 (95% CI: 1.198–4.748) for UEBMI, URBMI, NRCMS and other health insurance, respectively.

## 4. Discussion

In this nationally representative study in China, the prevalence of self-reported multimorbidity was 56.73% among the middle-aged and older Chinese population, which slightly differed from the prevalence reported in previous studies in low- and middle-income countries (LMIC) [9,25]. This prevalence was higher than Yao’s results [15], who also used data from CHARLS and found that multimorbidity occurred in 42.4% of the participants. The difference between the studies is largely due to heterogeneity in study populations and methodologies.

Multimorbidity prevalence differed by gender and socioeconomic status. Consistent with previous studies, women are more vulnerable to multimorbidity compared with men [10,26,27,28]. A study showed that female gender was associated with a 1.31-fold increased risk of multimorbidity [15]. This may be explained by the fact that women have a longer life expectancy than men, and the prevalence of multimorbidity increases with advancing age, especially in women. Moreover, it is noteworthy that Chinese women generally have less access to health resources than men, and may have poorer health consequently [15]. Furthermore, the prevalence of multimorbidity increased substantially with age but decreased among the oldest respondents. A possible explanation is that older adults tend to have impaired cognitive function and may underreport chronic diseases. Another explanation is that older respondents may be a selected sample with fewer chronic conditions [29]. Respondents with a Sishu/home school education had a higher possibility of developing multimorbidity. As highlighted in previous studies, education may play an important role in the development of preventive measures for multimorbidity [9]. Respondents with basic medical insurance, including UEBMI, URBMI and NRCMS, had a higher prevalence of multimorbidity than those with no health insurance. This can be explained by the fact that respondents with no health insurance had limited access to health resources and were liable to the underdiagnosis of chronic conditions [1,30].

Regarding multimorbidity patterns in this study, four patterns were identified: the *relatively healthy class*, *respiratory class*, *stomach-arthritis class* and *vascular class*. A study conducted in Spain with patients older than 65 years old identified six multimorbidity patterns: a *nonspecific pattern*, *musculoskeletal*, *endocrine-metabolic*, *digestive/digestive-respiratory*, *neurological* and *cardiovascular pattern* [27]. Zhang et al. identified five multimorbidity clusters among Chinese adults aged at least 60 years: a *relatively healthy class*, *respiratory class*, *vascular class*, *stomach-arthritis class* and *multisystem morbidity class* [1]. In the present study, 65.02% of the participants were classified into the relatively healthy class, which coincided with previous studies showing a high proportion of participants in the relatively healthy class, with 60.4% in Korea [31] and 63.8% in Spain [24]. Similar to studies in developed countries [24,31], we identified a *vascular class* (18.60%) characterized by the high prevalence of hypertension, dyslipidemia, stroke and heart disease. In line with previous studies [1], our study identified a *stomach-arthritis class*. Diseases of the digestive system and arthritis were often clustered among Chinese adults [14]. The explanation might be that there was a high incidence of stomach diseases and arthritis among Chinese adults and the medication for arthritis may have serious side effects for the stomach [15].

Regarding gender differences in multimorbidity patterns, we found a higher probability of belonging to the *stomach-arthritis class* among women. Zhang et al. [1] found that women had a higher probability of belonging to all multimorbidity patterns. This difference is largely due to different age restrictions. A recent meta-analysis [32] demonstrated that among adults aged 40 and over in China, the prevalence of arthritis for women was 20.50%, which was higher than among men (18.99%). Chinese women had higher chances of having arthritis than men.

In contrast to previous evidence showing that people with a low SES had a higher probability of displaying multimorbidity patterns related to cardio-metabolic conditions [24,33], our study found that respondents with a higher SES were more likely to be in the *vascular class*, including a middle school education and above, urban residence, higher consumption and medical insurance. The results were consistent with Zhang’s research conducted among adults aged 60 years and above [1]. This could be explained by the fact that respondents with a higher SES have a higher possibility of avoiding physically demanding tasks and consume high-calorie foods, which may lead to a high possibility of chronic diseases such as hypertension, dyslipidemia and diabetes. Another reason might be the underdiagnosis of chronic conditions for respondents in disadvantaged SES groups [1].

Lifestyle variables were found to be influencing factors for multimorbidity patterns. Compared to respondents who never consumed alcohol, drinkers were less likely to be in the *respiratory class* and *stomach-arthritis class*, given the possible health benefits of light and regular alcohol intake. Respondents with better health were more likely to consume alcohol. However, for smoking, past smoking and current smoking might increase the risk of being in the *stomach-arthritis class*. The more detailed measurement of alcohol consumption and smoking is necessary in future research. Because of our cross-sectional study design, we could not make causal inferences between drinking, smoking and multimorbidity patterns.

An interesting finding in this study is that we did not find an urban–rural difference in the prevalence of multimorbidity or in multimorbidity patterns. Furthermore, respondents with a Sishu/home school education and below had a higher risk of multimorbidity, but those with a middle school education and above were more likely to be in the *vascular class*. Previous studies indicated that the prevalence of multimorbidity was higher among rural older adults [20,34]. A study conducted in India found that participants residing in urban areas were more likely to report complex multimorbidity [35]. These inconsistencies may be partly due to the differences in sampling size and chronic conditions. In the present study, urban and more educated respondents were more likely to be in the *vascular class*. A previous study cited unhealthier lifestyles among urban and more educated residents, including westernized diets and sedentary habits, and greater exposure to environmental pollutants as risk factors for vascular disease [36]. In addition, urban and more educated respondents may have been less likely than their counterparts to engage in agricultural work.

A major strength of the present study is the analysis of a large, high-quality database, representative of a large population. However, some limitations must be taken into account, such as the cross-sectional design and the self-reporting of chronic diseases. The presence of chronic diseases was probably underreported, because only selected diagnoses were covered by the survey. Based on the cross-sectional design, this study cannot draw a causal relationship between socioeconomic characteristics and the prevalence and patterns of multimorbidity. A longitudinal study should be designed to validate the possible causal hypotheses in the future. Moreover, some important diseases with high prevalence among older people were not included in the study due to data availability, which may have resulted in the underestimation of the multimorbidity prevalence and misclassification of multimorbidity patterns.

## 5. Implications

The examination of gender and socioeconomic differences in multimorbidity patterns has great implications for clinical practice and health policy. The results may provide insights that could help to manage multimorbidity patients and improve health resource allocation. The characteristics of the high-risk groups identified in the study may help to develop and implement interventions to prevent the more serious consequences of multimorbidity [37]. Future health system development should move from preventing and controlling a single chronic condition to addressing the multimorbidity of older Chinese adults [20]. Continuing efforts to improve the understanding of multimorbidity among subgroups will help to develop healthcare models to address the specific needs of different subgroups.

## 6. Conclusions

In conclusion, the prevalence of multimorbidity was relatively high among Chinese middle-aged and older adults. The results showed that women, a lower education level and health insurance were associated with a higher prevalence of multimorbidity. Four specific multimorbidity patterns, the “*relatively healthy class*”, “*respiratory class*”, “*stomach-arthritis class*” and “*vascular class*”, were identified. Women were more likely to be in the *stomach-arthritis class*. Increased age and higher SES were positively associated with the *vascular class*.

## Figures and Tables

**Figure 1 ijerph-19-16956-f001:**
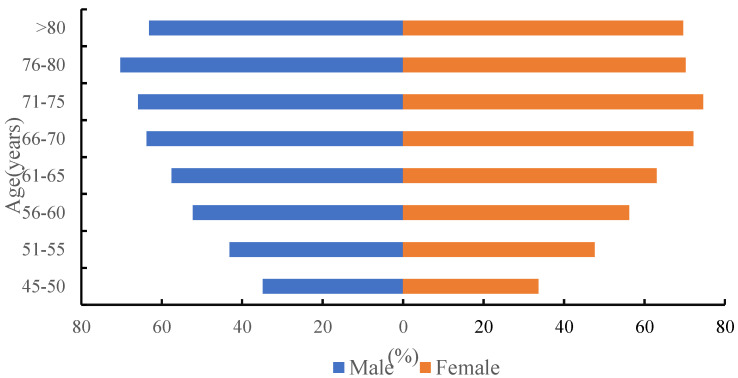
Weighted prevalence of multimorbidity by age and gender.

**Figure 2 ijerph-19-16956-f002:**
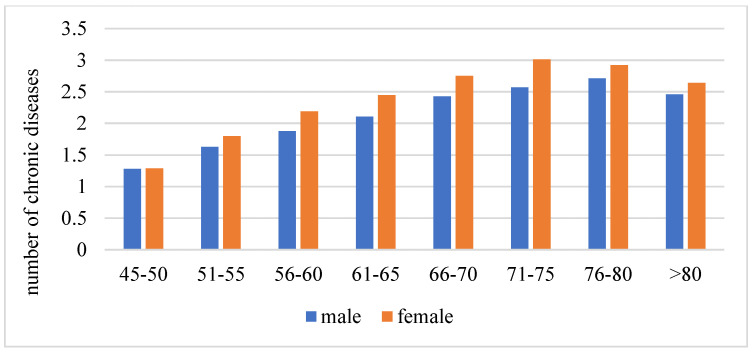
Number of chronic diseases by age and gender.

**Table 1 ijerph-19-16956-t001:** Characteristics of the sample (*n* = 19559).

Variables	Total (%)	Mean Number of Chronic Diseases (SD)	*p* Value	Proportion with Multimorbidity	*p* Value
All participants	19,559 (100%)	2.17 (1.88)		56.73%	
Gender					
Male	9307 (47.58%)	2.08 (1.84)	<0.001	54.65%	<0.001
Female	10,252 (52.42%)	2.26 (1.92)		58.71%	
Marriage status					
Married	16,619 (84.97%)	2.09 (1.84)	<0.001	55.20%	<0.001
Windowed	2512 (12.84%)	2.62 (2.01)		66.58%	
Others	428 (2.19%)	2.31 (2.20)		53.37%	
Education					
Sishu/home school and below	8537 (43.65%)	2.32 (1.93)	<0.001	60.49%	<0.001
Elementary school	4289 (21.93%)	2.14 (1.87)		54.99%	
Middle school	4258 (21.77%)	2.01 (1.81)		53.37%	
High school and above	2475 (12.65%)	2.06 (1.85)		54.18%	
Residence					
Rural	11,675 (59.69%)	2.15 (1.87)	0.16	56.21%	0.184
Urban	7884 (40.31%)	2.19 (1.90)		57.26%	
Alcohol consumption					
Never	12,899 (65.95%)	2.33 (1.95)	<0.001	60.44%	<0.001
Occasionally	1466 (7.50%)	1.99 (1.81)		50.83%	
Usually	5111 (26.13%)	1.88 (1.69)		50.66%	
Smoking status					
Never	11,211 (57.32%)	2.18 (1.88)	<0.001	56.93%	<0.001
Quit	445 (2.28%)	2.69 (2.04)		67.11%	
Current	7825 (40.01%)	2.16 (1.88)		56.55%	
Difficulty in ADL					
Yes	3615 (18.48%)	3.40 (2.10)	<0.001	81.72%	<0.001
No	15,944 (81.52%)	1.90 (1.72)		51.25%	
Self-rated health					
Less than good	15,066 (77.03%)	2.48 (1.93)	<0.001	64.43%	<0.001
Good	4493 (22.97%)	1.17 (1.30)		31.83%	
Health insurance					
None	583 (2.98%)	1.91 (1.83)	<0.001	48.96%	<0.001
UEBMI	2975 (15.21%)	2.32 (1.97)		60.63%	
URBMI	3189 (16.30%)	2.18 (1.97)		55.72%	
NRCMS	12,598 (64.41%)	2.11 (1.87)		56.20%	
Others	214 (1.09%)	1.91 (2.02)		49.79%	

Proportions with multimorbidity are weighted results adjusted for non-response of individuals and households. UEBMI: Urban Employee Basic Medical Insurance; URBMI: Urban Resident Basic Medical Insurance; NRCMS: New Rural Cooperative Medical Scheme; Others: private medical insurance and other medical insurance.

**Table 2 ijerph-19-16956-t002:** Multinomial logit analysis of the factors associated with multimorbidity patterns.

Variables	Relatively Health Class	Respiratory Class		Stomach-Arthritis Class		Vascular Class	
	RRR	RRR	95%CI	RRR	95%CI	RRR	95%CI
Gender (ref = Male)							
Female	1.00	0.795	0.501–1.261	1.595 ***	1.276–1.993	0.953	0.745–1.220
Age	1.00	1.038 ***	1.026–1.050	1.030 ***	1.023–1.038	1.033 ***	1.027–1.041
Marriage status (ref = Married)							
Windowed	1.00	0.946	0.733–1.220	0.966	0.798–1.170	0.921	0.778–1.090
Others	1.00	1.675 **	1.086–2.584	1.660 **	1.139–2.418	0.728	0.487–1.090
Education (ref = Sishu/home school and below)							
Elementary school	1.00	0.961	0.727–1.271	0.997	0.845–1.176	1.130	0.973–1.313
Middle school	1.00	0.958	0.667–1.376	0.869	0.712–1.061	1.363 ***	1.148–1.619
High school and above	1.00	0.627	0.403–0.977	1.103	0.834–1.459	1.401 ***	1.162–1.690
Residence (ref = Rural)							
Urban	1.00	1.162	0.885–1.526	0.897	0.776–1.038	1.438 ***	1.236–1.672
In PCE	1.00	1.001	0.915–1.096	1.095 **	1.026–1.168	1.114 ***	1.053–1.179
Alcohol consumption (ref = Never)							
Occasionally	1.00	0.604 **	0.438–0.832	0.793 *	0.612–1.027	0.781 *	0.613–0.995
Usually	1.00	0.779 *	0.597–1.017	0.794 **	0.650–0.970	0.894	0.747–1.070
Smoking status (ref = Never)							
Quit	1.00	1.120	0.599–2.090	1.581 *	0.996–2.509	1.053	0.672–1.649
Current	1.00	1.354	0.854–2.145	1.458 **	1.176–1.808	0.933	0.715–1.219
Difficulty in ADL (ref = No)							
Yes	1.00	2.148 ***	1.783–2.587	3.248 ***	2.819–3.743	1.986 ***	1.739–2.269
Self-rated health (ref = Less than good)							
Good	1.00	0.387 ***	0.249–0.603	0.175 ***	0.136–0.224	0.351 ***	0.297–0.415
Health insurance (ref = None)							
UEBMI	1.00	1.411	0.836–2.380	2.260 ***	1.456–3.506	3.200 ***	2.148–4.768
URBMI	1.00	1.259	0.774–2.047	1.704 **	1.134–2.561	2.353 ***	1.608–3.443
NRCMS	1.00	1.550 *	0.986–2.438	1.327	0.905–1.946	2.021 ***	1.403–2.912
Others	1.00	1.339	0.586–3.057	1.145	0.562–2.332	2.385 **	1.198–4.748

*** *p* < 0.01; ** *p* < 0.05; * *p* < 0.1.

## Data Availability

Publicly available datasets were analyzed in this study. These data can be found at http://charls.pku.edu.cn/ assessed on 8 March 2022.

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
