# Peer review of "Gender and Socioeconomic Differences in the Prevalence and Patterns of Multimorbidity among Middle-Aged and Older Adults in China"

_ijerph, 2022, doi:10.3390/ijerph192416956_

Round 1

Reviewer 1 Report

l   Since most readers are not familiar with China's social insurance system, when UEBMI, URBMI and NRCMS are mentioned for the first time, they should be fully stated and briefly introduced.

l   In 2.3. Statistical Analysis, the statistical method of mean number of chronic diseases in Table 1 was not mentioned, and the p-values of 2.08 (1.84) for males and 2.06 (1.92) for females were significant and may be incorrected. Please confirm the results of the analysis again.

l   Most of the Alcohol consumption items in Table 2 were statistically significant, but there is no mention of them in the text. Is it overlooked or are there other considerations?

l   In Line 226, the authors mentioned that "Lower education was associated with higher possibility for multimorbidity.", but according to the results in Table 2, the higher the education level, the higher the RRR in the Vascular class, and the descriptions were contradictory. The same problem existed in the conclusion. Please review again and make appropriate corrections.

l   Line 256-264, the author should search more literature on the relationship between SES and vascular-related diseases, which should be able to support the author's argument.

Author Response

1.  Since most readers are not familiar with China's social insurance system, when UEBMI, URBMI and NRCMS are mentioned for the first time, they should be fully stated and briefly introduced.

We added in page 3 “there were three main social medical insurance in China- urban employee basic medical insurance (UEBMI), urban resident basic medical insurance (URBMI) and new rural cooperative medical scheme (NRCMS)”

2. In 2.3. Statistical Analysis, the statistical method of mean number of chronic diseases in Table 1 was not mentioned, and the p-values of 2.08 (1.84) for males and 2.06 (1.92) for females were significant and may be incorrected. Please confirm the results of the analysis again.

We checked the results and changed in Table 1 and page 4 line 156 “The mean number of chronic diseases for all participants was 2.17, 2.08 for males and 2.26 for females respectively.”

3. Most of the Alcohol consumption items in Table 2 were statistically significant, but there is no mention of them in the text. Is it overlooked or are there other considerations?

We added the discussion about alcohol consumption from line 270 to 277.

4. In Line 226, the authors mentioned that "Lower education was associated with higher possibility for multimorbidity.", but according to the results in Table 2, the higher the education level, the higher the RRR in the Vascular class, and the descriptions were contradictory. The same problem existed in the conclusion. Please review again and make appropriate corrections.

We have done logistic regression (not show in the manuscript) and found respondents with elementary school and below had higher possibility for multimorbidity. In table 2, respondents with middle school and above were more likely to be in vascular class. For other multimorbidity patterns, we didn’t the association between education and multimorbidity patterns. We have made appropriate description in the manuscript.

5. Line 256-264, the author should search more literature on the relationship between SES and vascular-related diseases, which should be able to support the author's argument.

We have added in the discussion.

Reviewer 2 Report

1. The implication of the study should  be included in abstract

2. Why the specific Gender and Socioeconomic variables(education, residence , insurance )  are included in the study? A proper justification is missing

3. Why the Socioeconomic variables presented in  Table2 are missing from table-1 ( e.g  alcohol , smoking habit etc)?

4.  A separate subsections (i. e social implications & academic implications)   must included in discussion section to strengthen the implication of the study.

5. Detailed discussion of LCA will help the reader to understand the use of LCA 

Author Response

1. The implication of the study should be included in abstract.

We added in abstract “The examination of gender and socioeconomic differences for multimorbidity patterns had great implications for clinical practice and health policy. The results may provide insights to manage multimorbidity patients and improving health resources allocation.”

2. Why the specific Gender and Socioeconomic variables (education, residence, insurance) are included in the study? A proper justification is missing.

Previous studies showed inconsistent results in gender and socioeconomic difference in the prevalence and patters of multimorbidity. We also stated why choose these variables in page 3 from line 99 to line 120.

3. Why the Socioeconomic variables presented in Table2 are missing from table-1 (e.g  alcohol , smoking habit etc)?

We have added these variables in table 1.

4. A separate subsections (i. e social implications & academic implications)   must included in discussion section to strengthen the implication of the study.

We have added a separate subsection of implication in the section of discussion.

5. Detailed discussion of LCA will help the reader to understand the use of LCA 

We added in page 3 “LCA is based on structural equation modeling and is useful for determining subtypes of cases or groups in multivariate categorical data”. “The adjusted Bayesian Information Criterion (BIC) and the consistent Akaike Information Criterion (AIC) were used to determining the optimal number of latent classes. Based on the evaluation of a variety of model fit statistics, the present study examined two to six classes and selected the best fitting solution”.

Round 2

Reviewer 1 Report

Even though alcohol consumption is described in the discussion, it should also be described in the results section. The author believes that light alcohol intake can bring health benefits, but the RRR of usual Alcohol consumption is also lower than that of never, which does not conform to the author's discussion. The authors have to present more evidence to prove this association.

The author did logistic regression to prove that lower education was associated with higher possibility for multimorbidity, but did not describe this opposite result in either the results or the discussion. Authors must describe in the results, and state in the Discussion why the opposite results were found and what confounding factors might have been affected.

Author Response

     (1) Even though alcohol consumption is described in the discussion, it should also be described in the results section. The author believes that light alcohol intake can bring health benefits, but the RRR of usual Alcohol consumption is also lower than that of never, which does not conform to the author's discussion. The authors have to present more evidence to prove this association.

      We have described alcohol consumption in the results section. Please see line 158 to 160, line 198 to 202. We have made correct description in the discussion.

     (2)The author did logistic regression to prove that lower education was associated with higher possibility for multimorbidity, but did not describe this opposite result in either the results or the discussion. Authors must describe in the results, and state in the Discussion why the opposite results were found and what confounding factors might have been affected.

      We have added in the results and the discussion. We also stated the possible reason in the discussion from line 291 to 296.

Reviewer 2 Report

The comments are appropriately addressed, however additional explanation of the response is needed.

Author Response

      The comments are appropriately addressed, however additional explanation of the response is needed.

     We have done.

Round 3

Reviewer 1 Report

There are no further comments.

Author Response

There are no further comments.

We have read the manuscript again and made some minor revisions.
